

# pcr: an R package for quality assessment, analysis and testing of qPCR data

Mahmoud Ahmed and Deok Ryong Kim

Department of Biochemistry and Convergence Medical Sciences and Institute of Health Sciences, Gyeongsang National University School of Medicine, Jinju, Gyeongnam, South Korea

## ABSTRACT

**Background**. Real-time quantitative PCR (qPCR) is a broadly used technique in the biomedical research. Currently, few different analysis models are used to determine the quality of data and to quantify the mRNA level across the experimental conditions.

**Methods**. We developed an R package to implement methods for quality assessment, analysis and testing qPCR data for statistical significance. Double Delta $C_T$ and standard curve models were implemented to quantify the relative expression of target genes from $C_T$ in standard qPCR control-group experiments. In addition, calculation of amplification efficiency and curves from serial dilution qPCR experiments are used to assess the quality of the data. Finally, two-group testing and linear models were used to test for significance of the difference in expression control groups and conditions of interest.

**Results**. Using two datasets from qPCR experiments, we applied different quality assessment, analysis and statistical testing in the pcr package and compared the results to the original published articles. The final relative expression values from the different models, as well as the intermediary outputs, were checked against the expected results in the original papers and were found to be accurate and reliable.

**Conclusion**. The pcr package provides an intuitive and unified interface for its main functions to allow biologist to perform all necessary steps of qPCR analysis and produce graphs in a uniform way.

# INTRODUCTION

Real-time quantitative PCR (qPCR) is a commonly used technique to analyze the relative gene expression level in the biomedical research. In most cases, small scale experiments are designed to quantify the level of mRNA among experimental conditions. Some advanced machines and optimized protocols have simplified the experiments to a highly efficient one-step process, allowing the effective analysis of a large scale of qPCR data. However, all processes for assessing the quality of the data, performing the analysis and reporting the results are not done in the most uniform way across the literature (*Bustin & Nolan, 2004*). Different analysis models have been proposed and implemented in different software environments (*Pabinger et al., 2014*). Furthermore, requirements and guidelines for reporting qPCR data were independently introduced (*Bustin et al., 2009*).

Corresponding author
Deok Ryong Kim, drkim@gnu.ac.kr

**Table 1  Average $C_T$ value for c-myc and GAPDH at different input amounts.**

| Input RNA (ng) | c-myc average $C_T$ | GAPDH average $C_T$ | $\Delta C_T$ c-myc-GAPDH |
|---|---|---|---|
| 1.0 | $25.59 \pm 0.04$ | $22.64 \pm 0.03$ | $2.95 \pm 0.05$ |
| 0.5 | $26.77 \pm 0.09$ | $23.73 \pm 0.05$ | $3.04 \pm 0.10$ |
| 0.2 | $28.14 \pm 0.05$ | $25.12 \pm 0.10$ | $3.02 \pm 0.11$ |
| 0.1 | $29.18 \pm 0.13$ | $26.16 \pm 0.02$ | $3.01 \pm 0.13$ |
| 0.05 | $30.14 \pm 0.03$ | $27.17 \pm 0.06$ | $2.97 \pm 0.07$ |
| 0.02 | $31.44 \pm 0.16$ | $28.62 \pm 0.10$ | $2.82 \pm 0.19$ |
| 0.02 | $32.42 \pm 0.12$ | $29.45 \pm 0.08$ | $2.97 \pm 0.14$ |

In this report, we introduce an open source R package for performing quality assessment, modeling and testing for statistical significance of qPCR data in a uniform way. In its current version, the pcr package implement two methods for relative quantification of mRNA expression proposed originally by *Livak & Schmittgen (2001)*, in addition to the necessary steps to check the assumption of these methods. Also, we implement a number of methods to check for statistical significance in qPCR data which were introduced in SAS by *Yuan et al. (2006)*. Finally, the package provides unified interface to make the analysis accessible and the ability to make graphs of the different analysis steps for visual inspection and preparation of publication-level figures. We start by describing the process for generating the data in the original papers, briefly introduce the methods and apply them to the original data using the pcr.

## MATERIALS & METHODS

### Data sources

To illustrate the usage of the pcr package and to apply it to qPCR data, we used real qPCR datasets from two published papers. In addition, we compared the results obtained by the pcr package to that of the original paper to ensure the reliability. In their original article, *Livak & Schmittgen (2001)* collected total RNA from human tissues; brain and kidney. c-myc and GAPDH primers were then used for cDNA synthesis and used as input in the PCR reaction. Seven different dilutions where used as input to the PCR reaction (three replicates each), this dataset was referred to as ct3 and shown in Table 1. Six replicates for each tissue were run in separate tubes. This dataset was referred to as ct1 through this document and shown along with the difference calculations in Tables 2 and 3. At the second work, *Yuan et al. (2006)* extracted total RNA from *Arabidopsis thaliana* plant treated and control samples (24 samples each), and performed qPCR analyses using MT7 and ubiquitin primers. This dataset was referred to as ct4 and shown the results of the different testing methods that applied in the original paper in Table 4. In all datasets, the raw $C_T$ (Cycle Threshold), also known as $C_q$, was recorded (*Bustin et al., 2009*) and will be used as input to the pcr analysis pipeline.

**Table 2 Relative quantification using comparative ($\Delta\Delta C_T$) method (separate tubes).**

| Tissue | c-myc $C_T$ | GAPDH $C_T$ | $\Delta C_T$ c-myc-GAPDH | $\Delta\Delta C_T$ $\Delta C_T - \Delta C_{T,Brain}$ | c-myc$_N$ Rel. to Brain |
|---|---|---|---|---|---|
| Brain | 30.72 | 23.7 | | | |
| | 30.34 | 23.56 | | | |
| | 30.58 | 23.47 | | | |
| | 30.34 | 23.65 | | | |
| | 30.5 | 23.69 | | | |
| | 30.43 | 23.68 | | | |
| Average | $30.49 \pm 0.15$ | $23.63 \pm 0.09$ | $6.86 \pm 0.17$ | $0.00 \pm 0.17$ | 1.0 (0.9–1.1) |
| Kidney | 27.06 | 22.76 | | | |
| | 27.03 | 22.61 | | | |
| | 27.03 | 22.62 | | | |
| | 27.1 | 22.6 | | | |
| | 26.99 | 22.61 | | | |
| | 26.94 | 24.18 | | | |
| Average | $27.03 \pm 0.06$ | $22.66 \pm 0.08$ | $4.37 \pm 0.10$ | $-2.50 \pm 0.10$ | 5.6 (5.3–6.0) |

**Table 3 Relative quantification using the standard curve method (separate tube).**

| Tissue | c-myc (ng) | GAPDH (ng) | c-myc$_N$ norm. to GAPDH | c-myc$_N$ Rel. to Brain |
|---|---|---|---|---|
| Brain | 0.033 | 0.51 | | |
| | 0.043 | 0.56 | | |
| | 0.036 | 0.59 | | |
| | 0.043 | 0.53 | | |
| | 0.039 | 0.51 | | |
| | 0.040 | 0.52 | | |
| Average | $0.039 \pm 0.004$ | $0.54 \pm 0.034$ | $0.07 \pm 0.008$ | $1.0 \pm 0.12$ |
| Kidney | 0.40 | 0.96 | | |
| | 0.41 | 1.06 | | |
| | 0.41 | 1.05 | | |
| | 0.39 | 1.07 | | |
| | 0.42 | 1.06 | | |
| | 0.43 | 0.96 | | |
| Average | $0.41 \pm 0.016$ | $1.02 \pm 0.052$ | $0.40 \pm 0.025$ | $5.5 \pm 0.35$ |

**Table 4 Statistical significance using different testing methods.**

| Test | $\Delta\Delta C_T$ (estimate) | p-value | Confidence interval |
|---|---|---|---|
| Multiple regression | −0.6848 | <0.0001 | (−0.4435, −0.9262) |
| ANOVA | −0.6848 | <0.0001 | (−0.4435, −0.9262) |
| t-test | −0.6848 | <0.0001 | (−0.4147, −0.955) |
| Wilcoxon test | −0.6354 | <0.0001 | (−0.4227, −0.8805) |

## Statistical methods

In contrast with the absolute quantification of the amount of mRNA in a sample, the relative quantification uses a internal control (reference gene) and/or a control group (reference group) to quantify the mRNA of interest relative to these references. This relative quantification was sufficient to draw conclusions in most of the biomedical applications involving qPCR. A few methods were developed to perform these relative quantification. These methods generally require different assumptions and models for the analysis. The most common two of these methods were described here in the following sections.

### The comparative $C_T$ methods

The comparative $C_T$ methods assume that the cDNA templates of the gene/s of interest as well as the control/reference gene have similar amplification efficiency, and also that this amplification efficiency is near perfect. This means that, at a certain threshold during the linear portion of the PCR reaction, the amount of the gene of the interest and the control double each cycle. Another assumption is that the expression difference between two genes or two samples can be captured by subtracting one (gene or sample of interest) from another (reference). The final assumption is that the reference doesn't change with the treatment or the course in question. The formal derivation of the double delta $C_T$ model is described here *Livak & Schmittgen (2001)*. Briefly, the $\Delta\Delta C_T$ is given by:

$$\Delta\Delta C_T = \Delta C_{T,q} - \Delta C_{T,cb} \tag{1}$$

And the relative expression by:

$$2^{-\Delta\Delta C_T} \tag{2}$$

where $\Delta C_{T,q}$ is the difference in the $C_T$ (or their average) of a gene of interest and a reference gene in a group of interest. $\Delta C_{T,cb}$ is the the difference in the $C_T$ (or their average) of a gene of interest and a reference gene in a reference group. The error term is given by:

$$s = \sqrt{s_1^2 + s_2^2} \tag{3}$$

where $s_1$ is the standard deviation of a gene of interest and $s_2$ is the standard deviation of a reference gene.

### Standard curve

In comparison, this model doesn't assume perfect amplification but rather actively use the amplification in calculating the relative expression. Therefore, when the amplification efficiency of all genes are 100% both methods should give similar results. The standard curve method is applied using two steps. First, serial dilutions of the mRNAs from the samples of interest are used as input to the PCR reaction. The linear trend of the log input amount and the resulting $C_T$ values for each gene are used to calculate an intercept and a slope. Secondly, these intercepts and slopes are used to calculate the amounts of mRNA of the genes of interest and the control/reference in the samples of interest and the control sample/reference. These amounts are finally used to calculate the relative expression in a manner similar to the later method, just using division instead of subtraction. The formal derivation of the model is described here (*Yuan et al., 2006*). Briefly, the amount of RNA

in a sample is given by:

$$\text{log amount} = \frac{C_T - b}{m}.$$ (4)

And the relative expression is given by:

$$10^{\text{log amount}}$$ (5)

where $C_T$ is the cycle threshold of a gene. $b$ is the intercept of $C_T$ log10 input amount. $m$ is the slope of $C_T$. And the error term is given by:

$$s = (cv)(\bar{X})$$ (6)

where:

$$cv = \sqrt{cv_1^2 + cv_2^2}$$ (7)

where $s$ is the standard deviation. $\bar{X}$ is the average. $cv$ is the coefficient of variation or relative standard deviation.

### Statistical significance tests

Assuming that the assumptions of the first methods are holding true, the simple $t$-test can be used to test the significance of the difference between two conditions ($\Delta C_T$). $t$-test assumes, in addition, that the input $C_T$ values are normally distributed and the variance between conditions are comparable. Wilcoxon test can be used if sample size is small, and those two last assumptions are hard to achieve.

Two use the linear regression here. A null hypothesis is formulated as following,

$$C_{T,\text{target,treatment}} - C_{T,\text{control,treatment}} = C_{T,\text{target,control}} - C_{T,\text{control,control}}.$$ (8)

This is exactly the $\Delta\Delta C_T$ value as explained earlier. So the $\Delta\Delta C_T$ is estimated and the null is rejected when $\Delta\Delta C_T \neq 0$.

### Quality assessment

Fortunately, regardless of the method used in the analysis of qPCR data, The quality assessment can be done in a similar way. It requires an experiment similar to that of calculating the standard curve. Serial dilutions of the genes of interest and controls are used as input to the reaction and different calculations are made. The amplification efficiency is approximated be the linear trend between the difference between the $C_T$ value of a gene of interest and a control/reference ($\Delta C_T$) and the log input amount. This piece of information is required when using the $\Delta\Delta C_T$ model. Typically, the slope of the curve should be very small and the $R^2$ value should be very close to one. A value of the amplification efficiency itself is given by $10^{-1/\text{slope}}$ and should be close to 2. Other analysis methods are recommended when this is not the case. Similar curves are required for each gene using the $C_T$ value instead of the difference for applying the standard curve method. In this case, a separate slope and intercept for each gene are required for the calculation of the relative expression.
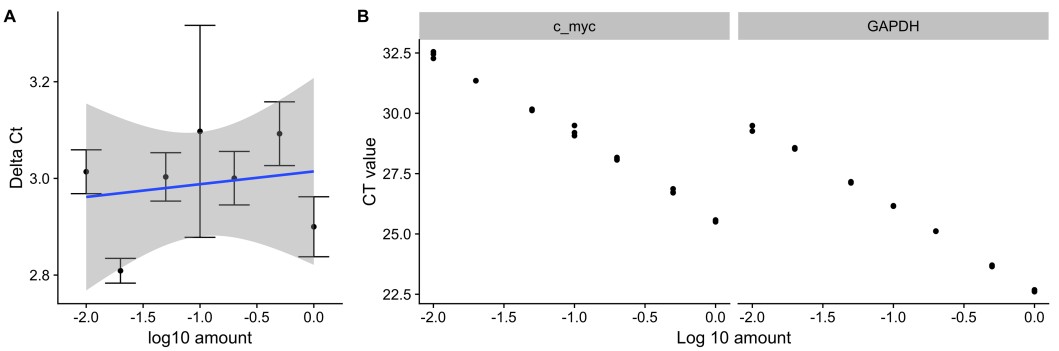

**Figure 1** **Amplification efficiency and standard curves of c-myc and GAPDH.** Seven different dilutions of RNA were used as an input to synthesize cDNA, then to a real-time quantitative PCR reaction with c-myc and GAPDH primers. (A) $\Delta C_T$ values were calculated by subtracting the $C_T$ values from three independent samples of the control gene(GAPDH) from the target c-myc. Averages and standard deviations are shown as points and error bars. The blue line represents the linear trend between the $\Delta C_T$ and log10 of the input amount. (B) $C_T$ values from three independent samples of c-myc and GAPDH are shown with the corresponding log10 input amounts.

## RESULTS & DISCUSSION

### Availability & installation

The pcr packages is available on CRAN, the main repository for R packages and can be installed by invoking **install.packages** in an R ($\geq$3.4.2) session. The package's source code is also available on github, https://github.com/MahShaaban/pcr along with the development version.

```
# install the pcr package from CRAN
install.packages('pcr')
```

The examples shown in this article are explained in greater details in the package vignette that can be accessed through browseVignette('pcr'). Moreover, the package documentation provides detailed instruction on the input and the output of each function (e.g., ?pcr_analyze).

### Functionality & user interface

The pcr package provides different methods for performing quality assessment, modeling and testing real-time qualitative PCR data through the unified interface of three functions: pcr_assess, pcr_analyze and pcr_test, respectively.

#### Quality assessment

pcr_assess provides two methods for assessing the quality of qPCR data. These are 'efficiency' and 'standard_curve' to calculate the amplification efficiency and gene standard curves as described in the methods section. The following code block applies both methods to the dataset ct3, shown in Table 1. Using the argument **plot** as TRUE in the pcr_assess function provides the a graphic presentation of the amplification and the standard curves as shown in Fig. 1.

```
# load required libraries
library(pcr)
library(ggplot2)
library(cowplot)
library(dplyr)
library(xtable)
library(readr)

# pcr_assess
## locate and read data
fl <- system.file('extdata', 'ct3.csv', package = 'pcr')
ct3 <- read_csv(fl)

## make a vector of RNA amounts
amount <- rep(c(1, .5, .2, .1, .05, .02, .01), each = 3)

## calculate amplification efficiency
res1 <- pcr_assess(ct3,
                   amount = amount,
                   reference_gene = 'GAPDH',
                   method = 'efficiency')

## calculate standard curves
res2 <- pcr_assess(ct3,
                   amount = amount,
                   method = 'standard_curve')

## retain curve information
intercept <- res2$intercept
slope <- res2$slope
```

### Analysis models

Similarly, pcr_analyze provides two methods to model the $C_T$ values and calculates the relative expression of target genes. 'delta_delta_ct' performs the $\Delta\Delta C_T$ method described previously. The average relative expression of the target gene in the condition of interest is given by the Eqs. (1) and (2) and the standard deviation by Eq. (3). The calculations are applied to the dataset 'ct1', shown in Table 2 and Fig. 2A. 'relative_curve' performs the relative standard curve quantification, average relative expression/amount of the target gene in the condition of interest is given by Eqs. (4) and (5) and the standard deviation by Eqs. (6) and (7). The calculation is applied to the same dataset 'ct1' and is shown in Table 3 and Fig. 2B.

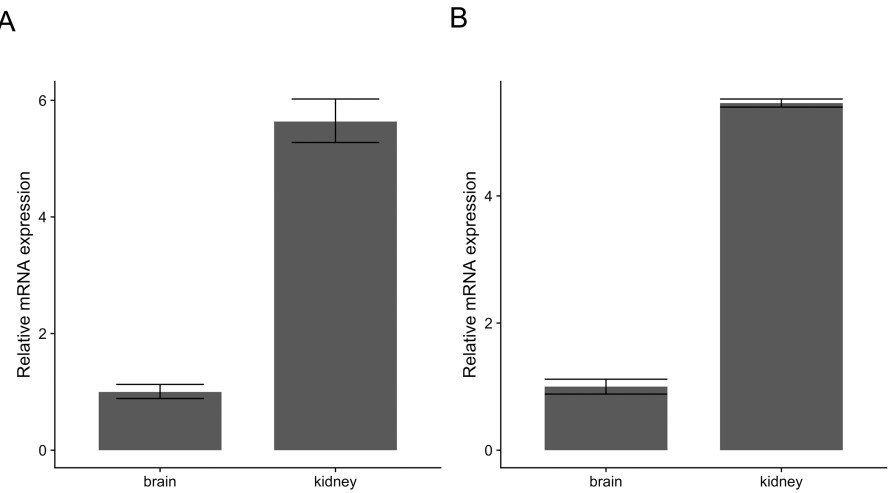

**Figure 2  Relative expression of c-myc in human brain and kidney tissues.** Total RNA from human brain and kidney tissues were used to synthesize cDNA and real-time quantitative PCR reaction with c-myc and GAPDH primers. $C_T$ values from six independent replicates were used to calculate the expression of c-myc in the kidney normalized by GAPDH and relative to the brain using The $\Delta\Delta C_T$ (A) and the standard curve methods (B). Averages and standard deviations are shown as bars and error bars.

```
# pcr_analyze
## locate and read raw ct data
fl <- system.file('extdata', 'ct1.csv', package = 'pcr')
ct1 <- read_csv(fl)

## add grouping variable
group_var <- rep(c('brain', 'kidney'), each = 6)

# calculate all values and errors in one step
## mode == 'separate_tube' default
res1 <- pcr_analyze(ct1,
                    group_var = group_var,
                    reference_gene = 'GAPDH',
                    reference_group = 'brain')

## calculate standard amounts and error
res2 <- pcr_analyze(ct1,
                    group_var = group_var,
                    reference_gene = 'GAPDH',
                    reference_group = 'brain',
                    intercept = intercept,
                    slope = slope,
                    method = 'relative_curve')
```

**Table 5  Different testing methods applied to the same dataset.**

|  | Gene | Estimate | p_value | Lower | Upper | Term |
|---|---|---|---|---|---|---|
| t.test | target | −0.68 | 0.00 | −0.96 | −0.41 | |
| wilcox.test | target | −0.64 | 0.00 | −0.88 | −0.42 | |
| lm | target | −0.68 | 0.00 | −0.95 | −0.41 | group_vartreatment |

### *Significance testing*

Finally, pcr_test can be used to calculate useful statistics, *p*-values and confidence intervals on the previous models. Two tests are available of the two-group comparisons; 't.test' and 'wilcox.test' to test for the difference of the normalized target gene expression ($\Delta C_T$) in one condition to another. Linear regression, 'lm', can be applied to estimate these differences between multiple conditions and a reference (Eq. (8)). The following code applies different testing methods to the dataset 'ct4'. The dataset was published original in (*Yuan et al., 2006*), along with results of different testing method (Table 4). Table 5 shows the results of the three different tests as implemented in pcr_test.

```
# pcr_test
# locate and read data
fl <- system.file('extdata', 'ct4.csv', package = 'pcr')
ct4 <- read_csv(fl)

# make group variable
group <- rep(c('control', 'treatment'), each = 12)

# test using t-test
tst1 <- pcr_test(ct4,
                 group_var = group,
                 reference_gene = 'ref',
                 reference_group = 'control',
                 test = 't.test')

# test using wilcox.test
tst2 <- pcr_test(ct4,
                 group_var = group,
                 reference_gene = 'ref',
                 reference_group = 'control',
                 test = 'wilcox.test')

# testing using lm
tst3 <- pcr_test(ct4,
                 group_var = group,
                 reference_gene = 'ref',
                 reference_group = 'control',
                 test = 'lm')
```

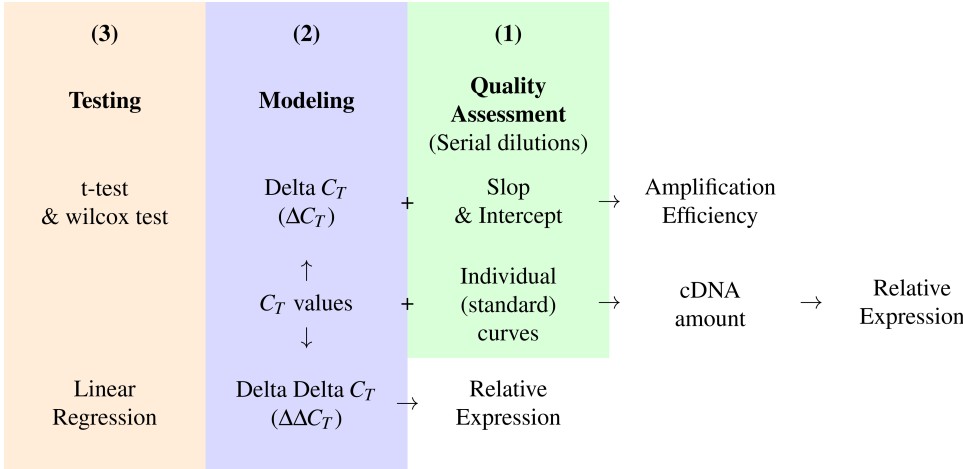

**Figure 3** **A conceptual workflow of the analysis of RT-qPCR data.** A graphic description of the input, intermediary values and final output of different steps of the analysis pipeline and their relations. Colors represent distinct steps and the numbers are the order in which they are applied in a typical analysis.

## The pcr package workflow

In Fig. 3, we suggest a workflow based on three steps; Quality assessment, modeling and testing for statistical significance, these can be applied by using the pcr package functions; pcr_assess, pcr_analyze and pcr_test respectively. The workflow is based on the $C_T$ from two kinds of experiments. First, the $C_T$ values for the genes of interest and a reference gene in the different experimental groups. Second, the $C_T$ values of the same genes from a serial dilution experiment. The amplification efficiency can be calculated by calculating the $\Delta C_T$ from the first and the slope and the intercept from the second experiment. The $\Delta\Delta C_T$ model and the relative expression are then calculated and tested. In the case of the two groups comparison, any of the testing methods can be applied otherwise the linear regression can be used for multiple group comparisons. Alternatively, the standard curve methods can be applied by calculating the amount of cDNA in each sample from the $C_T$ values and the individual curves for each gene. These amounts are finally used to obtain the relative expression of the genes of interest.

## Comparison with other packages

*Pabinger et al. (2014)* surveyed the tools used to analyze qPCR data across different platforms. They included nine R packages which provide very useful analysis and visualization methods. Table 6 shows a comparison of the features of these nine packages from the original publication, in addition to the pcr package. Some of these packages focuses one certain model and some are designed to handle high-throughput qPCR data (e.g., chipPCR). Most of these packages are hosted in CRAN and a few on the Bioconductor so they adhere to Bioconductor methods and data containers (e.g., qpcR).

In comparison, pcr provides a unified interface for different quality assessment, analysis and testing models. The different functions take similar inputs to perform different analysis steps which requires minimal formating at the end of the user. The main input and the

**Table 6  Comparison with available R packages for qPCR quantification.**

| Package | Raw[a] | Efficiency | Abs quant | Rel quant | Error prop. | Norm. | NA handling | Graphs | Stats. | MIQE[b] |
|---|---|---|---|---|---|---|---|---|---|---|
| chipPCR | + | + | | | | | + | + | | + |
| ddCT | | | | + | | + | | + | + | + |
| dpcR | | | + | | | | + | + | + | + |
| EasyqpcR | | + | | + | | + | + | | | + |
| HTqPCR | | | | + | | + | + | + | + | |
| FPK-PCR | + | + | | | | | | | | |
| NormqPCR | | | | + | | + | + | | | + |
| qpcR | + | + | + | + | + | + | + | + | | + |
| qpcrNorm | | | | | | + | | + | + | |
| pcr (proposed) | | + | | + | + | + | | + | + | + |

**Notes.**
[a]Calculates the $C_T$ values from raw florescence data.
[b]Complies with MIQE (Minimum Information for Publication of Quantitative Real-Time PCR Experiments) recommendations for reporting RT-qPCR results.

output are tidy **data.frame**, and the package source code follows the tidyverse practices. This package targets the small scale qPCR experimental data and the R users. The interface and documentation choices were made with such users in mind and require no deep knowledge in specific data structures or complex statistical models. Users can go from the raw $C_T$ values through different analysis steps to publication graphs in a few simple lines of code.

### Limitations & future directions

The current version of the pcr package (1.1.0) provides only methods to estimate the expression of genes in a certain condition relative to another. Other methods were proposed for absolute quantification of the copy number of genes in samples (*Whelan, Russell & Whelan, 2003*). Also, the comparative $C_T$ methods assume that the PCR reaction has a close to perfect amplification rates. Other methods were proposed to model the data when this assumption is not true (*Liu & Saint, 2002*; *Tichopad et al., 2003*). We are planning to implement methods for absolute quantification and dealing with less than perfect amplification efficiency cases in a future version of the package.

## CONCLUSION

To sum, the pcr package is an open source R package for quality assessing, modeling and testing real-time quantitative PCR data. The package provide an intuitive and unified interface for its main functions to allow biologist to perform all necessary steps of qPCR analysis and produce graphs in a uniform way.

## ACKNOWLEDGEMENTS

We thank all lab members for the critical discussion at the development of this R package.

### Funding

This work was supported by the Basic Science Research Program through the National Research Foundation of Korea funded by the Ministry of Education (2015R1D1A01019753) and by the Ministry of Science, ICT and Future Planning (NRF-2015R1A5A2008833). The funders had no role in study design, data collection and analysis, decision to publish, or preparation of the manuscript.

### Grant Disclosures

The following grant information was disclosed by the authors:
National Research Foundation of Korea: 2015R1D1A01019753.
Ministry of Science: NRF-2015R1A5A2008833.

### Competing Interests

The authors declare there are no competing interests.

### Author Contributions

- Mahmoud Ahmed conceived and designed the experiments, performed the experiments, analyzed the data, contributed reagents/materials/analysis tools, prepared figures and/or tables, authored or reviewed drafts of the paper, approved the final draft.
- Deok Ryong Kim conceived and designed the experiments, prepared figures and/or tables, authored or reviewed drafts of the paper, approved the final draft.

### Data Availability

Github: https://github.com/MahShaaban/pcr.

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
