# Peer review of "pcr: an R package for quality assessment, analysis and testing of qPCR data"

_PeerJ, doi:10.7717/peerj.4473_

## Round 0.1 · original submission · Major Revisions

I agree with Reviewer 1, that the motivation and the knowledge gap is not clearly described. In addition to saying, that PCR provides a ‘unified interface’; it would be helpful to provide concrete examples of how this package is more comprehensive/ user-friendly. Also, please provide a point-wise response to the reviewer’s comments and my comments below.

Since there are several functions which compose of the workflow, it would be helpful to present all available functions as in infographic, describing the entire flow of the package.

In addition, unless it is absolutely required, please remove the strict dependence on R (>= 3.4.2). This prevents users with slightly older R version, from using and testing this package.

You have mentioned Pabinger et al. provides an overview of all available packages. However, you can use the background and discussion, to provide more examples as to how this package improves upon these existing nine packages. Often readers find a table describing various features useful in choosing a package for their data. In this case, focusing on type of input and output (data.frame/S4), available methods for normalization, visualization could be some of the example features to compare across packages.

Reviewer 1 ·

Basic reporting

- The manuscript clearly summarizes the functionality of the presented R-package and outlines basic use cases. The background section may be shortened as these methods have been extensively reviewed (as also cited) elsewhere (please see http://relative.gene-quantification.info/, http://gene-quantification.com/hkg.html#reviews)

- Line 49 - please improve grammar

Experimental design

- The relevance of the package is not clearly presented. Many R-packages with much greater functionality (also including the functions presented in the manuscript) are available and in my view this package does not fill any “knowledge gap”. Already published R-packages include: https://cran.r-project.org/web/packages/qpcR/index.html, https://bioconductor.org/packages/release/bioc/html/EasyqpcR.html, https://bmcgenomics.biomedcentral.com/articles/10.1186/1471-2164-13-296, …

- The authors argue that a new unified interface is provided, but this interface still requires a certain input format.

- The authors may include more data preprocessing methods to directly import data from different qPCR vendors.

- The package is clearly outlined, also the vignette presents many details

Validity of the findings

- The provided sample data allows users to explore the included package functionality

- Authors should respect the MIQE guidelines (https://www.ncbi.nlm.nih.gov/pubmed/19246619), including their proposed naming convention

Reviewer 2 ·

Basic reporting

Article is written in clear, structured and professional manuscript schema with appropriate background, rationale, methods, examples, and importantly limitations. Authors have throughout adhered to most commonly used best-practices for developing and documenting R package.

Experimental design

Authors presented a R package to include two of commonly used statistical approaches in interpreting PCR data with an easy-to-understand concepts behind each of statistical framework for new users. Authors have explained in detail core concept, biological context and limitation of each of these two methods before describing implementation of those in their R package.

Validity of the findings

Authors have made source code available on CRAN and github and did an impressive job of adhering to best-practices for developing and documenting R package. Authors provide both visual and tabulated form of data for interpreting PCR data for both irrespective of one's expertise in statistics. I have a few minor comments or rather suggestions in interpreting results:

1. For comparative Ct method, besides assuming similar amplification efficiency for reference and gene of interest, a resulting Ct value typically above 40-45 (depending upon calibration of machine) may not be reliable and should be flagged accordingly, i.e., Ct value for gene of interest beyond 40-45 cycle is non-deterministic as that gene may not be expressed at all or would have very minimal expression.

2. For t-test, unless pcr_test function inherits t-test arguments, I suggest to add an option for user to provide either "one-tailed" or "two-tailed" t-test depending of an experimental design.

Additional comments

Authors did an impressive job at structuring and writing this article in easy-to-understood language but yet providing details on interpreting PCR data with both biological and statistical context. I second and upvote their suggestions in limitations to improve this package and also suggests to add R-Shiny interface to analyze PCR data in interactive manner.

Thanks!

---

## Round 0.2 · Minor Revisions

We have now received a response from both reviewers, and your revision have been received well. However, reviewer 1 has indicated several important points regarding adding a limitations section to better orient the audience. Also, please provide a point-wise response to the reviewer 1’s comments.

Reviewer 1 ·

Basic reporting

- Background section: The authors reply that they want to present a complete and stand-alone document for readers. Unfortunately, the section is not complete and might not stand on its own. I still would suggest to shorten it.

- I would very much recommend adding a section that clarifies where your method is applicable and where it is not. Since it only covers a very small part of the analysis workflow, potential users need to be informed what scenarios and data they can analyze with the package.

Experimental design

- Concerning data format: Can the authors reason, why they did not use the AnnotatedDataFrame? (https://www.rdocumentation.org/packages/Biobase/versions/2.32.0/topics/AnnotatedDataFrame) There is even a package for reading: ReadqPCR (https://www.rdocumentation.org/packages/ReadqPCR/versions/1.18.0/topics/read.qPCR)

- The authors reply that “input data are always the raw ct values”. Can they please elaborate (also in the manuscript) why raw Ct values are taken for statistical testing when a reference gene is used?

- How will the authors include new preprocessing methods to directly import data from different qPCR vendors in their package?

Validity of the findings

- I think by adding a short statement that Cq will be used instead of Ct no reader will be confused. (MIQE has been cited over 6000 times.

- Novel diagram: Why does the diagram start with statistical testing (on the left side). What is the input? What is the output?

- Table: How does “pcr” perform error propagation? How does it comply with the MIQE guidelines?

- Spelling mistakes: line 236

Reviewer 2 ·

Basic reporting

no comments

Experimental design

Thank you for consideration of comment. In my knowledge, there are no set rules for cycle threshold from method perspective. In theory, depending upon sensitivity of PCR machine and type of experimental design with proper negative controls, it can detect even the minute levels of transcript in the lysate, i.e. at much higher cycle threshold than 40-45. However, in practical scenario, e.g., at Ct value of 45, for tissue-specific expression of a given gene transcripts, it is questionable if that transcript is expressed at a minute but functional level or tissue does not express that isoform at all. I like the idea of providing warning message with base R function but leave the decision on authors' end if they do not find compelling reference to support such argument.

Related references about cycle threshold but not regarding formal cut-off:
1. https://www.qiagen.com/mx/resources/faq?id=783d4566-9ad9-4fab-9936-182beda65617&lang=en
2. Schmittgen TD, Livak KJ. Analyzing real-time PCR data by the comparative C(T) method. Nat. Protoc. 2008;3:1101–8. PMID: 18546601

Validity of the findings

no comments

---

## Round 0.3 · accepted · Accept

There are however a few minor issues which should be addressed before the final publication.

As reviewer 1 also pointed out the figure adds value, but intuitively it would be great if it can be read from left to right or any other direction with arrows indicating flow of the data. You may consider adding a statement in vignette/manuscript, regarding the input being ct (cycle threshold)/ cq (cycle quantification), citing Bustin et. al.